# Reply to: Pitfalls in using phenanthroline to study the causal relationship between promoter nucleosome acetylation and transcription

Benjamin J. E. Martin [1] & LeAnn J. Howe [2✉]

**REPLYING TO** S. Zencir et al. *Nature Communications* https://doi.org/10.1038/s41467-022-30350-3 (2022)

We appreciate the comments of Zencir et al. regarding our manuscript on the impact of transcription on genome-wide histone acetylation patterns[1]. We agree that our paper shifts our understanding of how acetylation is targeted to transcribed genes and thus the more scrutiny the results receive, the better. With this being said, we respectfully disagree that the loss of histone acetylation in cells treated with the transcription inhibitor, 1,10-phenanthroline (1,10-pt), is due to the disruption of histone acetyltransferase (HAT) binding to promoters. Although we do not dispute that the binding of some transcription activators, and thus HATs, may be impacted by this drug, the NuA4/piccolo HAT complex subunit, Epl1, is not lost at most promoters following 1,10-pt treatment. Using DESeq2's binomial test[2] with a lenient cutoff (*p*-value < 0.1, 1.5 fold-change), we observe a statistically significant loss of Epl1 binding at only 3.6% (198/5542) of promoters (Fig. 1a). This re-analysis of our data is completely consistent with the heatmap in our original manuscript (Figure S5A)[1], which shows that upon transcription inhibition: (1) Epl1 binding is stable at most promoters, albeit with a small fraction showing decreased binding, and (2) Epl1 is ubiquitously lost from 5′ gene body regions, consistent with our interpretation of transcription-dependent targeting of Epl1 to gene bodies. While the binding of Epl1 to both promoters and gene bodies is difficult to differentiate in the gene-dense yeast genome, the contrasting impact of Epl1 mutation and 1,10-pt treatment on Epl1 binding to these features confirms the specificity of our ChIP-seq experiments and strengthens our finding that Epl1 binding is stable to 1,10-pt at the vast majority of promoters. While Zencir et al. raise good points about the pitfals of metaplots, the heatmaps we included in our original paper address many of the points they now claim to uncover. Despite the limited impact of 1,10-pt on Epl1 occupancy, we observed loss of H4K8ac and H4K12ac at the majority of adjacent +1 nucleosomes (Fig. 1b, c). We do observe Epl1 loss at a larger portion of promoters with Epl1-peaks (126/562 promoters,

Fig. 1d), but Fig. 1e, f show that acetylation is lost at nucleosomes adjacent to all promoters in this class, even those exhibiting increased Epl1 occupancy, suggesting that loss of acetylation cannot be simply explained by loss of HAT recruitment. These data provide indirect support that the binding of transcription factors to most promoters is undisturbed upon 1,10-pt treatment, consistent with the results of Poramba-Liyanage et al., which characterized the impact of 1,10-pt on the occupancy of general transcription factors at a transcribed locus[3]. Interestingly, Fig. 1a, b, d in Zencir et al. also reinforce this point. At the genes shown, Rpb3 binding to promoters, presumably via recruitment by transcription activators, is not impacted by 1,10-pt. Regardless, the key point is that Epl1 occupancy at most genes is largely unaffected by 1,10-pt, which is inconsistent with the Zencir et al. statement that this drug has profound genome-wide effects on the localization and activity of transcription activators, as well as HAT and HDAC complexes. Again, we do not dispute that some promoters might be impacted, such as those of genes encoding ribosomal proteins and proteins involved in ribosome biogenesis, however, although these genes, represent a large portion of RNAPII initiation events, they are a small fraction of total genes (~350 in total), and thus account for a very small percentage of acetylated histones in the cell. Zencir et al. also speculate that increased targeting of the HDAC, Rpd3, could explain the decrease in acetylation following 1,10-pt treatment. However, we tested this possibility in our original paper (Supplementary Fig. 2e)[1], finding that the decrease in bulk histone acetylation was not due to increased HDAC activity.

A key issue that requires addressing is how Zencir et al. came to such radically disparate conclusions than us after analyzing our data. We believe that the answer lies in differences in how the data were analyzed. Processing and presenting genomic data require a number of decisions, such as what data to include, how to annotate specific genic regions, what statistical analyses to perform, and how to present figures. We respectfully disagree

[1] Department of Biological Chemistry and Molecular Pharmacology, Harvard Medical School, BCMP, LHRRB Building, Room 201a, 240 Longwood Ave., Boston, MA 02115, USA. [2] Department of Biochemistry and Molecular Biology, Life Sciences Centre, University of British Columbia, 2350 Health Sciences Mall, Vancouver, B.C. V6T 1Z3, Canada. ✉email: ljhowe@mail.ubc.ca

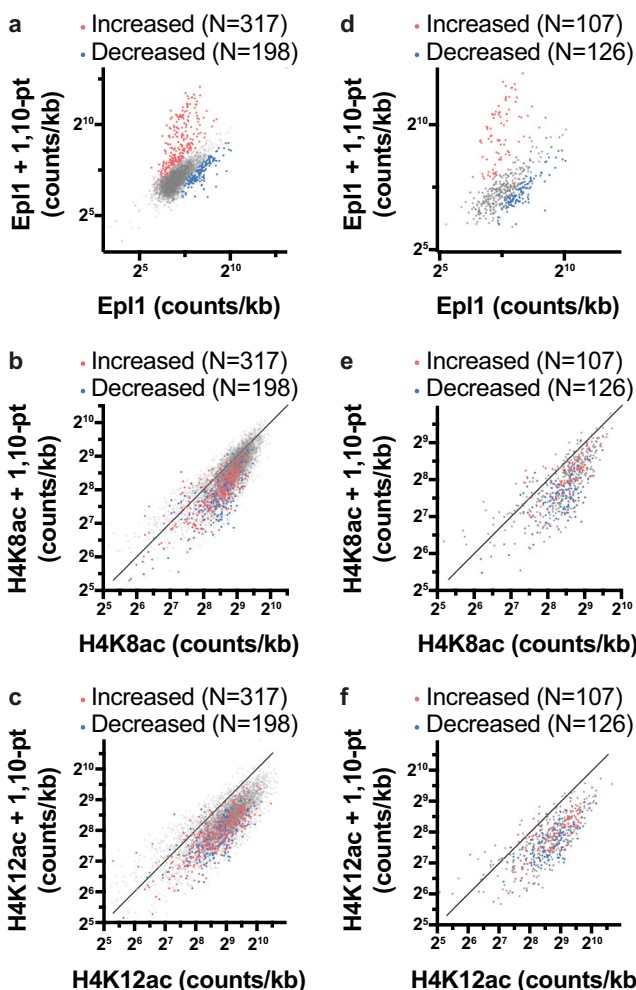

**Fig. 1 Transcription inhibition by 1,10-phenanthroline (1,10-pt) does not impact Epl1 occupancy at most promoters.** Scatter plots comparing the effect of 1,10-pt on fragment midpoint counts from Epl1 (**a**, **d**), H4K8ac (**b**, **e**), or H4K12ac (**c**, **f**) ChIP-seq, over promoters (**a** and **d**) or associated +1 nucleosomes (**b**, **c**, **e**, and **f**) of all (5542) genes (**a**, **b**, **c**) or of 562 genes with NDR Epl1 peaks identified in MNase ChIP-seq in control cells (**d**, **e**, **f**). Promoters with >1.5 fold-change in Epl1 using an adjusted *p*-value < 0.1 (DESeq2's binomial test[2]) are shown in red (increased) or blue (decreased). In **b**, **c**, **e**, and **f**, the line is: $y = x + 0$.

with many of the decisions made by Zencir et al. First, they based much of their argument that 1,10-pt treatment mirrors the effects of heat shock on the transcription of the top 400 expressed genes. However, previous work has shown that heat shock affects expression of only 10–15% of genes[4], while 1,10-pt achieves near universal transcription repression (Figure S3 in our original paper)[1]. As Zencir et al. highlight, the highly expressed ribosomal protein genes are repressed by heat shock and thus will make up a large proportion of the 400 genes they analyzed. Analysis of all genes reveals the flaws of their comparison, as 1,10-pt effects on RNAPII occupancy bear little resemblance to the effects of heat shock on RNAPII serine 5 phosphorylation[5] (Fig. 2a). Principle component analysis further underscores the differential effects of 1,10-pt treatment and heat shock on RNAPII occupancy (Fig. 2b). Second, Zencir et al. defined promoters as the 400 bp regions upstream of transcription start sites (TSSs), which is flawed for multiple reasons. Most gene TSSs are found ~30–40 bp within the first genic nucleosome[6] and thus, including regions immediately upstream of the TSS will measure Epl1 bound to these nucleosomes

as opposed to promoters. Also, yeast promoters feature pervasive divergent coding and non-coding transcription[7–9] and the average NDR width of yeast promoters is ~164 bp[6]. Thus, regions 400 bp upstream of TSSs will capture not just promoters, but a significant portion of transcribed regions as well. To highlight this point, Fig. 2c shows that regions 400 bp upstream of TSSs show comparable NET-seq read counts as those 400 bp downstream of TSSs. Multiple studies, including ours, have shown the transcription-dependent targeting of NuA4/Tip60 complexes to gene bodies[10, 11], and thus all signals originating from transcribed regions should be excluded when analyzing the requirement of transcription for Epl1 binding to promoters. For Fig. 1a, we defined promoters as the nucleosome depleted regions upstream of protein-coding genes with annotated TSSs. Other concerns we have with the analysis of our data by Zencir et al. are (1) overplotting of data in Fig. 1f, which masks the limited changes in Epl1 occupancies at "All promoters", (2) manipulating the Y-axes scales in Fig. 2d to enhance the difference of Epl1 loss at "Other" promoters, (3) lack of statistical analysis in Fig. 2d, and (4) comparison of loss of H3, not H4, acetylation to loss of Rpb1 Ser5p, not Rpb1, at not all, but a subset, of genes in Fig. 1e.

While we welcomed the request to reexamine our data generated from cells treated with 1,10-pt, it is important to note that experiments using 1,10-pt are not the only data supporting the transcription-dependence of histone acetylation in our manuscript[1]. Additional evidence of this in our original manuscript[1] is the directionality of transcription (Figs. 1g, h, s4) and the lack of colocalization of HATs with acetylation (Figures 3a, b, c, and d). Admittedly, these are correlative analyses, but we cannot envisage why chromatin-bound HATs would not robustly acetylate nucleosomes, especially those upstream of promoters, unless post-recruitment regulation was involved. While we agree with Zencir et al. that additional experiments mapping HATs and histone acetylation following transcription inhibition would be helpful, we argue that to a large degree these experiments have already been performed. Namely, in mESCs, inhibition of transcription initiation or elongation disrupts the binding of the mammalian homolog of Esa1, Tip60, to gene bodies, which mirrors the impact of 1,10-pt treatment in yeast[11]. Furthermore, groups using a RNAPII-degron or triptolide to inhibit transcription initiation observed a strong transcription-dependence of H3K27ac in human cell lines[12, 13]. In our hands, a Rpb2-degron approach revealed close to a 50% reduction in H4K8ac, despite the tendency of this approach to achieve incomplete depletion of transcribing RNAPII[12]. Thus, multiple groups, using varying methods to block transcription in diverse systems, all observed remarkably similar results. We believe that this is very strong evidence for the generality of our findings, that transcription does indeed shape histone acetylation. Finally, it should be noted that although our work conflicts with the widely accepted model of histone acetylation functioning upstream of transcription, it is compatible with data upon which this model is based. These include four key findings: (1) the interaction of HATs with transcription activators, (2) the requirement of HATs for full gene activation, (3) the packaging of active genes with acetylated histones, and (4) the weakening of chromatin structure by histone acetylation. All of these findings are consistent with a new model based on transcription-stimulation of HAT activity. In this model, HATs, regardless of their mode of targeting, function as components of feed-forward loops that reinforce the active transcription state through acetylating histones.

## Methods

FASTQ files were mapped as described in Martin et al. BAM files were filtered for 100–500 bp reads and total read midpoints overlapping nucleosome depleted regions[4] or +1 nucleosomes[14], were calculated using the Java Genomics Toolkit (https://github.com/timpalpant/java-genomics-toolkit) ngs.IntervalStats script. For DESeq2 analysis[2], size factors accounting for sequencing depth and the scaling factors reported in Martin et al.[1] were used. For calling NDRs with increased or decreased Epl1 binding, we used lenient cutoffs of a fold change of >1.5 and an

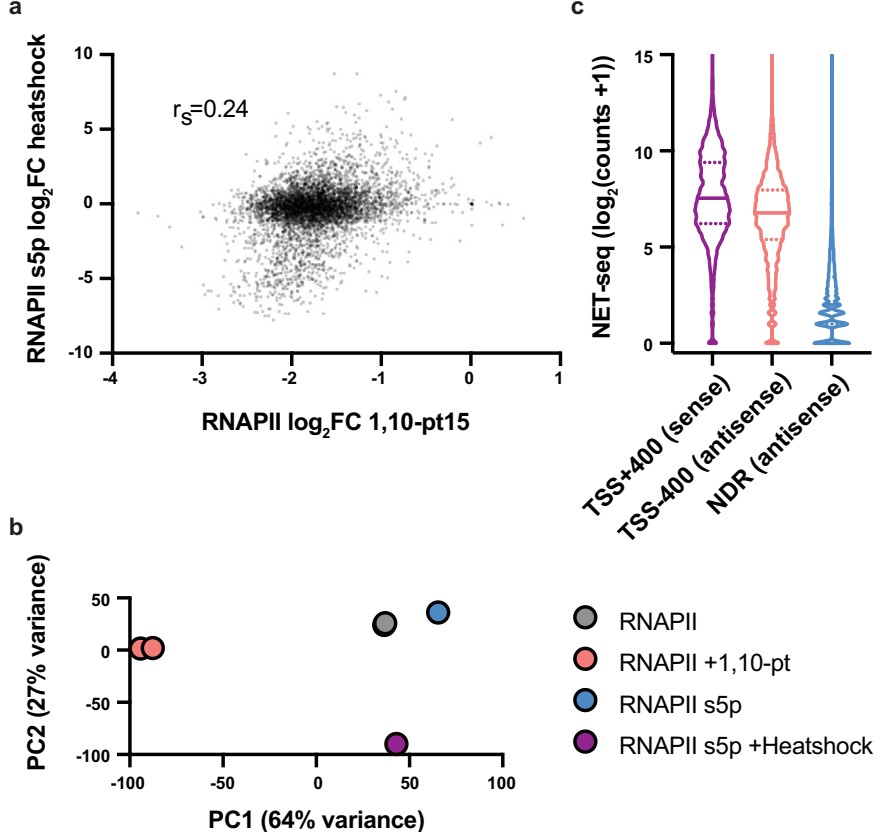

**Fig. 2 Transcription inhibition with 1,10-phenanthroline (1,10-pt) does not mirror the effect of heat shock on RNAPII occupancy at most genes.**
**a** Scatter plot comparing the $\log_2$ fold change (FC) in RNAPII occupancy following treatment with 1,10-pt with RNAPII serine 5 phosphorylation (s5p) following heat shock in gene 5′ regions (TSS to +400 bp) for all (5542) genes. **b** PCA plot showing the first two principal components calculated across TSS to +400 bp for all (5542) genes. **c** NET-seq 3′ read counts from TSS to +400 nt (sense), TSS to −400 nt (antisense), or in NDRs (antisense) depicted by violin plots for all (5542) genes.

adjusted $p$-value (FDR) of <0.1. For NET-seq data[15], processed bedGraphs were downloaded from GSM1673641 and GSM1673642. Sense and antisense read counts in specified windows were calculated using the Java Genomics Toolkit, summed across replicates, and violin plots made in Prism. For RNAPII serine 5 phosphorylation ChIP-seq[5], FASTQ files were downloaded from the SRA (SRR8450263 and SRR8450261) and mapped using BWA. Fragment centers, estimated using an assumed fragment length of 150 bp, were counted from the TSS to +400 bp then depth normalized to compare to RNAPII normalized as described in Martin et al. Principal component analysis (PCA) was performed on all 5542 genes (TSS+400) using the regularized log transformation and plotPCA functions of DEseq2[2].

**Reporting summary**. Further information on research design is available in the Nature Research Reporting Summary linked to this article.

## Data availability
Published datasets analyzed for this article include "GSE110287", "GSE110286", "GSE68484", and "GSE125226".

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

## Acknowledgements

Support for this work was provided by grants to L.J.H. from the Canadian Institutes of Health Research (PJT-162253) and Natural Sciences and Engineering Research Council (RGPIN-2018-04907). B.J.E.M. is supported by a Banting Postdoctoral Fellowship from the Canadian Institutes of Health Research.

## Author contributions

Data generation: B.J.E.M. Drafting and revision: B.J.E.M. and L.J.H.

## Competing interests

The authors declare no competing interests.
