## [Peer Review File · Nature Communications]

Reply to: Pitfalls in the use of phenanthroline to study the causal relationship between promoter nucleosome acetylation and transcriptionReviewers' Comments:

Reviewer #1:

Remarks to the Author:

The original paper from Martin et al. made the provocative suggestion that the majority of histone acetylation in yeast cells is targeted by active transcription, reversing the usual cause and effect model. Many, but not all, of the experiments involved using the chemical 1,10-phenanthroline to inhibit transcription. Histone acetylation dropped at both promoters and downstream regions upon 1,10-pt treatment, likely by reducing recruitment via RNA pol II, but not disrupting the recruitment of NuA4 HAT (assayed via the Epl1 subunit) to promoters by sequence-specific factors. Other treatments to inhibit transcription had a similar effect, but the analysis of these other conditions was not very extensive.

The correspondence from Zencir et al. questions whether the 1,10-pt effects can be directly and solely assigned to transcription inhibition, or whether other mechanisms might be responsible. They make a convincing case that 1,10-pt triggers a stress response, and use some data from the Martin paper to show that Epl1 binding was in fact changing at a significant fraction of bound promoters. The changes correlate with the particular transcription activator used at the promoters, suggesting that 1,10-pt is affecting specific signaling pathways rather than working through a general effect. Overall, I think this analysis raises important questions and is something that readers of the original Martin paper should take into consideration.

One issue I do have with the Zabnir analysis, which was also raised in the reply from Martin and Howe, is that they analyze only H3K27ac data to make their arguments about general histone acetylation. However, the Martin paper is primarily about NuA4 (Epl1), which is an H4 acetyltransferase. It would seem prudent to analyze the NuA4-linked H4 acetylation data to be sure the stress- and growth-related gene expression changes cited in support of their argument are also reflected in H4 acetylation.

In the reply from Martin and Howe, they concede that Epl1 binding does change at some promoters (their Fig 2a is strikingly similar to Fig 2a in Zabnir), but suggest that isn't sufficient to account for the loss of histone acetylation, particularly because 90% of promoters don't have Epl1 peaks (and therefore show no difference in binding with 1,10-pt), but still show a drop in acetylation (Fig 1a). They argue that the differing interpretations are at least partly due to differing definitions of promoter regions. I'm not convinced that's a major issue, and I note that in responding to my original review, the authors indicated that the majority of acetylation is not occurring at promoters anyway. Overall, I think the reply makes a reasonable defense their interpretation of data from original paper. Readers can consider both points of view and come to their own conclusions about which they think is most likely.

One comment about using the fact that TSA inhibition of HDACs did not diminish the response to 1,10-pt as support for the transcription-dependence model. I didn't pick up on this before, but it appears that deletion of Rpd3 (see Fig S2C), and to some extent Hda1, does significantly abrogate the acetylation drop caused by 1,10-pt transcription inhibition. This would seem to undercut the TSA argument.

Together, the two letters demonstrate how science is supposed to work. Opinions can differ, even when looking at the same data. Let's hope future experiments will weigh in to help the field come to a consensus.

Reviewer #2:

Remarks to the Author:

Having read the Martin et al. paper, the Comment by Zencir et al., and this Response, my comments are as follows:

Zencir et al. provide convincing evidence that treatment with 1,10-phenanthroline ("1,10-pt") causes a stress response that results in induction of many stress response genes and repression of other genes, most notably ribosomal protein genes and ribosome biogenesis genes. This effect was not considered in the manuscript by Martin et al., which is certainly a deficiency. I also agree with Zencir's criticism of the use of line graphs that average data for all genes or subsets of genes in the Martin et al. manuscript, as this can obscure effects seen at a limited number of genes or result in heterogeneous behavior averaging out to a net zero effect, among other problems. However, it is not clear to this reviewer that it invalidates entirely the conclusions in the original paper. In particular, the heat maps of Supplemental Figure 3 show that the large majority of genes show loss of Pol II and accompanying loss of H3K23ac, H4K8ac, and H4K12ac; the number of upregulated genes (showing Pol II gain) appears small, consistent with the relatively small number of genes indicated as being induced by 1,10-pt by Zencir et al. Supplemental Figure 5 also is convincing in showing that Epl1 association appears to decrease only modestly at the large majority of genes after 1,10-pt treatment. It seems likely that the RP genes, which are evidently repressed by 1,10-pt treatment and which have Rap1-bound promoters, are at the bottom of the heat map in Supplementary Figure 5 of Martin et al., representing a relatively small, though not important, cohort.

Martin et al., in their response, address the issue of how such disparate conclusions were arrived at by the two parties from analysis of the same data. The argument that defining promoters in this instance as the region 400 bp upstream of TSS's, as Zencir et al. have done in their analyses, is inappropriate has some merit. I'm not necessarily convinced that this accounts for all the difference. I also agreed with the general criticism from Zencir et al. that the novel idea (albeit supported by some additional studies) that activity of promoter-bound HATs is regulated post-recruitment deserves rigorous testing, and that experiments using anchor away or rapid degradation to rapidly abrogate transcription would have been useful, particularly since they allow a stronger control than in the 1,10-pt experiments, namely administration of auxin (or rapamycin) to untagged strains, as in Supplemental Figure 2b of Martin et al.

Martin et al. also object to the way in which some of the figures in the Comment by Zencir et al. are presented; I found these to be relatively minor objections as the figures in Zencir in general seemed, if sometimes somewhat idiosyncratic, to support their arguments in a fair manner. The exception to this might be Figure 2D, where a case could be made that the varied scales in the y-axis misrepresent the results. Even there, I am not convinced that rescaling so all results were presented with the same axis would affect interpretation.

Two reservations to the original Martin et al. manuscript that occurred to me that were not expressed by Martin et al. were 1) the possibility of nucleosome turnover affecting the degree of modified histones at genes was not considered and 2) loss of Epl1 at gene bodies, clearly seen, could affect acetylation at regions upstream. With regard to this latter point, the "-1" nucleosome at the region showing essentially no loss of Epl1 exhibited relatively modest decreases in histone acetylation (Supplemental Figure 2 of Martin et al.), and clearly some acetylation was retained (Figure 2f-h), so statements that transcription is required for acetylation of 5' nucleosomes are too strong in the original manuscript. Another unmentioned (minor) criticism of the original manuscript is that the Kaplan algorithm for nucleosome stability/positioning has been pretty convincingly refuted (Tillo and Hughes, 2009) (and others).

I fear this review has rambled a bit; my overall feeling is that the objections raised by Zencir et al. deserve publication, as does the response from Martin et al. These are both excellent groups of researchers disagreeing over what to some extent are technical issues, and the discussion will in my view interest a fairly wide group of researchers interested in genomic analyses. The Commentary by Zencir et al. also sheds light on weaknesses in the original Martin et al. paper that temper some of the conclusions drawn there. I can easily envision the manuscript and commentary being a productive topic in many graduate journal clubs.

Minor comment: For Figure 3 in the response to be meaningful, divergent promoters should be

excluded.

Tillo, D., and Hughes, T.R. (2009). G+C content dominates intrinsic nucleosome occupancy. *BMC Bioinformatics* 10, 442.

Reviewer #3:

Remarks to the Author:

Zencir make a convincing and valid point about the potential indirect effect of 1,10-pt, thereby making any of Martin's conclusions drawn from it equivocal. The Martin Reply makes an incorrect assertion about Zencir ("is due to the disruption of ...). Zencir do not argue that the results are due to indirect effects, but only that the possibility cannot be excluded. Zencir et al make a compelling case that 1,10-pt is a broadly acting inducer of stress in yeast. Interpretations of Martin et al experiments are based on the assumption that 1,10-pt is a specific inhibitor of RNAPII. Based on these tenets, we can join the Martin and Zencir claims in that a general stress response inhibits RNAPII, which other studies have shown. It is known that RNAPII inhibition persists for 5-15 min, followed by a gradual reactivation. 1,10-pt was used by Martin for 15 min. The crux of the problem is whether the set of experiments reported by Martin et al can really make a definitive statement about whether histone acetylation is a cause or a consequence of transcription. As Zencir point out 1,10-pt might have indirect effects, including genome-wide reprogramming of transcription and/or the signaling cascade brought on by this stress, leading to loss of acetylation by any number of mechanisms.

Martin et al and their Reply assume that NuA4 (Epl1) binds the vast majority of promoters or gene bodies (thereby justifying the use of 5,542 promoter in their denominator for reporting the fraction of genes losing Epl1 in the presence of 1,10-pt). However, no evidence was presented that shows Epl1 binding to this approximate number of promoters. The assumption of Epl1 detection at most promoters/gene bodies may be statistically no different from background. Therefore, any changes in such background signal (i.e., unsubstantiated Epl1 signal) is meaningless. The statement in Martin's Reply: "the key point is that Epl1 occupancy is largely unaffected by 1,10-pt..." is therefore invalid.

Both Zencir and Martin agree that there is considerable Epl1 loss (on average) where Epl1 binding can be detected. However, Zencir make the point that gene averaging obscures heterogeneous effects where some gene show loss of binding and others show no change or increased occupancy. This is a common and well know problem with gene averaging, and therefore problematic with the Martin study. For example, Fig 1 of Martin shows a gene average of 5206 genes, but as an average it really reflects only those genes that have the most RNAPII to lose (i.e., highly expressed genes) or the most acetylation to lose, and is not actually representative of all genes.

The Martin Reply indicates that even genes that gain Epl1 with 1,10-pt treatment lose the corresponding acetylation. Although the effect is modest, as shown in Fig. 2 of the Reply and in prior publications, the result is consistent with Martin's conclusion that HATs can be recruited to at least some promoters without them being active for acetylation. However, the potential for indirect effects does not allow an unequivocal claim. The results are consistent with their model, but is not unequivocal.

The Martin Reply indicates that Zencir's use of a 400 bp promoter window is flawed. However, the complex being examined here (NuA4) binds to promoters with large NDRs, including RP and other induced genes. This is clearly shown elsewhere when examining other NuA4 subunits (see yeastepigenome.org). As long as these genes are the focus of the analysis, 400 bp is appropriate. Most other genes have small promoters (NFRs), but there is no evidence of Epl1 binding there.

The Martin Reply states that Zencir Fig. 2D is plotted to locally expand the y-axis that enhances the apparent loss of Epl1. The Martin Reply is correct. All y-axes should start at zero. Further, the

analyses in both studies generally suffer from a lack of variance reporting in the metaplots.

As a final note, transcription is a cycle. So, inhibition of one part of the cycle (Pol II trxn) results in loss of histone acetylation. As a cycle, loss of histone acetylation could be temporally downstream of Pol II transcription, but also upstream the next transcription cycle. But even if 1,10-pt were acting directly and exclusively on RNAPII, and causation of histone acetylation by transcription is established, I don't think this tells us anything definitive about the role of histone acetylation. Transcription might cause histone acetylation and this acetylation might create a chromatin environment that promotes the next round of transcription. So it is a chicken-and-egg question applied to the transcription cycle that is not resolved here.

Reviewer #4:

Remarks to the Author:

Reply to: Pitfalls in the use of phenanthroline to study the causal relationship between promoter nucleosome acetylation and transcription

In their original manuscript, the authors showed that transcription inhibition upon treatment of 1,10 phenanthroline (1,10-pt) had no effect on NuA4 binding to promoter and proposed that histone acetylation is enhanced by RNA pol II transcription rather than promotes RNA pol II recruitment.

Although the authors did not mention a small number of genes showing differential Epl1 binding or those activated by 1,10-pt treatment in the original paper, I generally agree with the data presented in this manuscript and the original paper. In addition, the authors' concerns about the data provided by Albert et al. are also acceptable.

Reviewer #5:

Remarks to the Author:

In the Matters Arising, Albert and colleagues raise two important issues that argue against key conclusions from Martin et al. 2021, studying the cause-and-effect relationship between transcription and histone acetylation. The points raised by Albert are:

1. Martin et al. used the drug 1,10 phenanthroline which inhibits RNA Pol II transcription, to show that 15 min of treatment to yeast cells results in the loss of histone acetylation from gene promoters and gene bodies, as well as having effects on the binding of enzymes (HATs) that catalyze histone acetylation. However, Albert raises the point that acute treatment of 1,10 phenanthroline has been shown to affect signaling pathways, transcription factor binding, and induces a heat shock-like stress response. Therefore, the loss of histone acetylation is unlikely to be direct consequence of transcription inhibition by 1,10 phenanthroline.
2. Martin et al. used ChIP-seq to show that 1,10 phenanthroline results in an increase in the binding of a histone H4-specific HAT component Epl1 at a subset of gene promoters, despite the loss of histone acetylation. Based on this, they argued that promoter-bound HATs are unable to acetylate histones in the absence of transcription. Albert and colleagues reanalyzed the Epl1 ChIP-seq data to show that increased binding is limited to a smaller subset of promoters, while most of the promoters analyzed by Martin et al. actually show a loss of Epl1 binding. Mostly based on this analysis, Albert and colleagues argue that loss of histone acetylation upon 1,10 phenanthroline is rather due to lack of HAT binding or recruitment.

In the rebuttal to the Matter Arising, Martin and Howe argue that careful analysis of the ChIP-seq data with strictly (and more appropriately) defined promoter boundaries show both increase and decrease in Epl1 binding upon 1,10 phenanthroline, while histone acetylation is mainly decreased in both cases. They therefore conclude that loss of acetylation cannot be simply explained by loss of HAT

recruitment. Martin and Howe's response to this issue is convincing, and highlights the importance of carefully defining genomic regions for data analysis.

However, this argument over HAT binding is restricted to about 1/10th of all transcriptionally active yeast gene promoters at which Epl1 ChIP-seq shows factor binding in absence of the drug; and is therefore a minor/secondary point. Regardless of Epl1 binding by ChIP-seq, 1,10 phenanthroline results in decreased histone H4 acetylation at promoter-proximal +1 nucleosomes at the majority of yeast genes (not just a subset), as highlighted in the rebuttal. This suggests that Epl1 ChIP-seq cannot be treated as a reliable metric for HAT recruitment or activity. Nevertheless, Martin and Howe focused on mainly addressing this issue about HAT recruitment in their response.

The more important question raised by Albert and colleagues is whether the loss of histone acetylation upon 1,10 phenanthroline is a direct consequence of transcription inhibition. They point out that this can be better addressed with acute depletion of RNAPolII using an anchor-away or a degron-based approach. Martin et al. have used a RNAPolII subunit Rpb2 degron in their original study, but limited their analysis to bulk measurements of histone acetylation levels by immunoblotting. Therefore, the genome-wide effects of directly depleting/directly inhibiting RNAPolII on histone acetylation remains unknown. Martin and Howe point to a more recent study showing similar results in human cells using the TFIIF inhibitor Triptolide (which again does not directly inhibit RNA PolII). The rebuttal however demands a more compelling argument in support of a causal role of transcription/RNAPolII for nucleosomal histone acetylation.

In summary, Albert and colleagues have raised very important points to question some key conclusions from a seminal study which addresses a fundamental knowledge-gap in gene regulation. I recommend that this exchange is published with minor revision of the Martin and Howe response to better address the question of causality.

Responses to Reviewers' Comments

Reviewer #1

The original paper from Martin et al. made the provocative suggestion that the majority of histone acetylation in yeast cells is targeted by active transcription, reversing the usual cause and effect model. Many, but not all, of the experiments involved using the chemical 1,10-phenanthroline to inhibit transcription. Histone acetylation dropped at both promoters and downstream regions upon 1,10-pt treatment, likely by reducing recruitment via RNA pol II, but not disrupting the recruitment of NuA4 HAT (assayed via the Epl1 subunit) to promoters by sequence-specific factors. Other treatments to inhibit transcription had a similar effect, but the analysis of these other conditions was not very extensive.

The correspondence from Zencir et al. questions whether the 1,10-pt effects can be directly and solely assigned to transcription inhibition, or whether other mechanisms might be responsible. They make a convincing case that 1,10-pt triggers a stress response, and use some data from the Martin paper to show that Epl1 binding was in fact changing at a significant fraction of bound promoters. The changes correlate with the particular transcription activator used at the promoters, suggesting that 1,10-pt is affecting specific signaling pathways rather than working through a general effect. Overall, I think this analysis raises important questions and is something that readers of the original Martin paper should take into consideration.

One issue I do have with the Zabnir analysis, which was also raised in the reply from Martin and Howe, is that they analyze only H3K27ac data to make their arguments about general histone acetylation. However, the Martin paper is primarily about NuA4 (Epl1), which is an H4 acetyl-transferase. It would seem prudent to analyze the NuA4-linked H4 acetylation data to be sure the stress- and growth-related gene expression changes cited in support of their argument are also reflected in H4 acetylation.

In the reply from Martin and Howe, they concede that Epl1 binding does change at some promoters (their Fig 2a is strikingly similar to Fig 2a in Zabnir), but suggest that isn't sufficient to account for the loss of histone acetylation, particularly because 90% of promoters don't have Epl1 peaks (and therefore show no difference in binding with 1,10-pt), but still show a drop in acetylation (Fig 1a). They argue that the differing interpretations are at least partly due to differing definitions of promoter regions. I'm not convinced that's a major issue, and I note that in responding to my original review, the authors indicated that the majority of acetylation is not occurring at promoters anyway. Overall, I think the reply makes a reasonable defense their interpretation of data from original paper. Readers can consider both points of view and come to their own conclusions about which they think is most likely.

One comment about using the fact that TSA inhibition of HDACs did not diminish the response to 1,10-pt as support for the transcription-dependence model. I didn't pick up on this before, but it appears that deletion of Rpd3 (see Fig S2C), and to some extent

Hda1, does significantly abrogate the acetylation drop caused by 1,10-pt transcription inhibition. This would seem to undercut the TSA argument.

This reviewer did not overlook an issue when evaluating our original manuscript. Supplementary Figures 2c and d in the published manuscript show HDAC activity is required for removal of histone acetylation following 1,10-pt treatment. Since HDACs are required for loss of acetylation, these results cannot be used to determine whether acetylation loss in 1,10-pt-treated cells is due to reduced HAT activity or increased HDAC activity. The experimental result shown in Supplementary Figure 2e, addresses this question. This data is from cells first treated with 1,10-pt and then TSA. If 1,10-pt triggers loss of acetylation by increasing HDAC recruitment, then acetylation should increase when TSA is removed, which it does not.

Together, the two letters demonstrate how science is supposed to work. Opinions can differ, even when looking at the same data. Let's hope future experiments will weigh in to help the field come to a consensus.

We agree with this remark and such studies have already been published. Indeed, an article in a recent issue of *Nature Genetics* cited our work by stating "for lysine acetylation, this result mirrors elegant complementary experiments focused on histone acetylation in yeast" (PMID 35273399).

Reviewer #2

Having read the Martin et al. paper, the Comment by Zencir et al., and this Response, my comments are as follows:

Zencir et al. provide convincing evidence that treatment with 1,10-phenanthroline ("1,10-pt") causes a stress response that results in induction of many stress response genes and repression of other genes, most notably ribosomal protein genes and ribosome biogenesis genes. This effect was not considered in the manuscript by Martin et al., which is certainly a deficiency. I also agree with Zencir's criticism of the use of line graphs that average data for all genes or subsets of genes in the Martin et al. manuscript, as this can obscure effects seen at a limited number of genes or result in heterogeneous behavior averaging out to a net zero effect, among other problems. However, it is not clear to this reviewer that it invalidates entirely the conclusions in the original paper. In particular, the heat maps of Supplemental Figure 3 show that the large majority of genes show loss of Pol II and accompanying loss of H3K23ac, H4K8ac, and H4K12ac; the number of upregulated genes (showing Pol II gain) appears small, consistent with the relatively small number of genes indicated as being induced by 1,10-pt by Zencir et al. Supplemental Figure 5 also is convincing in showing that Epl1 association appears to decrease only modestly at the large majority of genes after 1,10-pt treatment. It seems likely that the RP genes, which are evidently repressed by 1,10-pt treatment and which have Rap1-bound promoters, are at the bottom of the heat map in

Supplementary Figure 5 of Martin et al., representing a relatively small, though not important, cohort.

Martin et al., in their response, address the issue of how such disparate conclusions were arrived at by the two parties from analysis of the same data. The argument that defining promoters in this instance as the region 400 bp upstream of TSS's, as Zencir et al. have done in their analyses, is inappropriate has some merit. I'm not necessarily convinced that this accounts for all the difference. I also agreed with the general criticism from Zencir et al. that the novel idea (albeit supported by some additional studies) that activity of promoter-bound HATs is regulated post-recruitment deserves rigorous testing, and that experiments using anchor away or rapid degradation to rapidly abrogate transcription would have been useful, particularly since they allow a stronger control than in the 1,10-pt experiments, namely administration of auxin (or rapamycin) to untagged strains, as in Supplemental Figure 2b of Martin et al.

While more data is generally always better, as outlined in our revised document, to a large degree these suggested experiments have already been performed and the results published. Further, although 1,10-phenanthroline may indirectly alter expression of a few hundred genes through triggering a stress response as opposed to directly inhibiting RNAPII activity, it should be noted that anchor away and auxin-induced degradation of RNAPII also trigger a "transcriptional stress response" resulting in upregulation of ~70 genes (PMID 27578267). While it seems counterintuitive that genes would be upregulated following RNAPII depletion, others have shown chromatin-bound RNAPII can be resistant to degradation (PMID 34678064) and thus one cannot rule out the significant presence of continuously transcribing RNAPII following either anchor away or auxin-induced degradation. Clearly, a lack of suitable approaches for inhibiting transcription in *S. cerevisiae* is one of the weaknesses of working with this organism, but we argue the rapid action of 1,10-phenanthroline, makes it the most suitable approach currently available.

Martin et al. also object to the way in which some of the figures in the Comment by Zencir et al. are presented; I found these to be relatively minor objections as the figures in Zencir in general seemed, if sometimes somewhat idiosyncratic, to support their arguments in a fair manner. The exception to this might be Figure 2D, where a case could be made that the varied scales in the y-axis mis-represent the results. Even there, I am not convinced that rescaling so all results were presented with the same axis would affect interpretation.

Two reservations to the original Martin et al. manuscript that occurred to me that were not expressed by Martin et al. were 1) the possibility of nucleosome turnover affecting the degree of modified histones at genes was not considered....

First, we would like to make an objection here. My understanding of this process is that the reviewers would be asked to weigh in on the critiques made by Zencir et al. and our response to them. I disagree that this should be used as an opportunity to subject our manuscript to yet another round of peer review. With this being said, we did consider

histone turnover as an explanation for the transcription-dependence of histone acetylation. There were, however, two reasons we did not follow-up on this line of investigation. First, the majority of histone turnover occurs at the nucleosomes adjacent to nucleosome depleted regions (PMID 17347438, 17679090), whereas acetylation is primarily found on the first 2-4 nucleosomes and only in the direction of transcription (see Supplementary Figure 4 in our published manuscript). As such, histone turnover shows poor correlation with histone acetylation. Second, we found that deletion of *ASF1*, a gene required for histone turnover in yeast, did not impact histone acetylation levels observed by immunoblot.

and 2) loss of Epl1 at gene bodies, clearly seen, could affect acetylation at regions upstream. With regard to this latter point, the “-1” nucleosome at the region showing essentially no loss of Epl1 exhibited relatively modest decreases in histone acetylation (Supplemental Figure 2 of Martin et al.), and clearly some acetylation was retained (Figure 2f-h), so statements that transcription is required for acetylation of 5' nucleosomes are too strong in the original manuscript.

We are unfortunately somewhat confused by the above comments. First, Supplemental Figure 2 is strictly immunoblot data, so we assume that this reviewer is referring to Figure 2. Second, we cannot envisage a mechanism for how loss of Epl1 over gene bodies could impact acetylation of regions upstream. Third, we are unsure how this relates to statements about acetylation of 5' nucleosomes. We do agree that Figure 2h, which shows histone acetylation of genes lacking divergent transcription, does indeed show a modest level of histone acetylation at the “-1 nucleosome”, and this acetylation does not disappear entirely following 1,10-pt treatment. However, the relative amount of acetylation is strikingly low considering that this is where the greatest occupancy of Epl1 is observed (Figure 2b).

Another unmentioned (minor) criticism of the original manuscript is that the Kaplan algorithm for nucleosome stability/positioning has been pretty convincingly refuted (Tillo and Hughes, 2009) (and others).

We thank this reviewer for bringing this to our attention but argue that it does not impact the findings of our study. Although Tillo and Hughes developed a simpler means to predict nucleosome occupancy based on sequence alone, genome-wide nucleosome occupancies predicted using both models show a 0.86 Pearson correlation coefficient. As such, in regard to this study, use of the Tillo and Hughes approach would have unlikely impacted the results.

I fear this review has rambled a bit; my overall feeling is that the objections raised by Zencir et al. deserve publication, as does the response from Martin et al. These are both excellent groups of researchers disagreeing over what to some extent are technical issues, and the discussion will in my view interest a fairly wide group of researchers interested in genomic analyses. The Commentary by Zencir et al. also sheds light on weaknesses in the original Martin et al. paper that temper some of the

conclusions drawn there. I can easily envision the manuscript and commentary being a productive topic in many graduate journal clubs.

We are especially grateful for this comment and love the idea that this discourse will be used to develop the next generation of scientists!

Minor comment: For Figure 3 in the response to be meaningful, divergent promoters should be excluded.

In Figure 3 we included divergent promoters to mirror the analysis of Zencir et al. It is clear from this Figure that defining promoters as 400 bp upstream of TSSs will inadvertently include many transcribed regions, likely due to inclusion of the 5' ends of divergently transcribed genes.

Tillo, D., and Hughes, T.R. (2009). G+C content dominates intrinsic nucleosome occupancy. *BMC Bioinformatics* 10, 442.

Reviewer #3

Zencir make a convincing and valid point about the potential indirect effect of 1,10-pt, thereby making any of Martin's conclusions drawn from it equivocal. The Martin Reply makes an incorrect assertion about Zencir ("is due to the disruption of ...). Zencir do not argue that the results are due to indirect effects, but only that the possibility cannot be excluded. Zencir et al make a compelling case that 1,10-pt is a broadly acting inducer of stress in yeast. Interpretations of Martin et al experiments are based on the assumption that 1,10-pt is a specific inhibitor of RNAPII. Based on these tenets, we can join the Martin and Zencir claims in that a general stress response inhibits RNAPII, which other studies have shown. It is known that RNAPII inhibition persists for 5-15 min, followed by a gradual reactivation. 1,10-pt was used by Martin for 15 min. The crux of the problem is whether the set of experiments reported by Martin et al can really make a definitive statement about whether histone acetylation is a cause or a consequence of transcription. As Zencir point out 1,10-pt might have indirect effects, including genome-wide reprogramming of transcription and/or the signaling cascade brought on by this stress, leading to loss of acetylation by any number of mechanisms.

Again, we do not disagree that 1,10-pt may alter transcription of some genes by triggering a stress response as opposed to inhibiting RNAPII activity. However, this effect can only explain our results if 1,10-pt triggers a global loss of transcription factor occupancy at thousands of promoters. Zencir et al. speculate that the binding of Iyh1 and Sfp1 to the promoters of ribosomal protein genes is impacted by 1,10-pt, but this represents only a few hundred genes. However, at the genes shown in Figures 1a, b, and d in Zencir et al., Rpb3 binding to promoters, presumably via recruitment by transcription activators, is not impacted by 1,10-pt. Further, the work of Poramba-Liyanage et al., 2020 showed that Reb1, one of the three "general regulatory" transcription factors in yeast, remains associated with a reporter locus following 1,10-pt

treatment. To further underscore that 1,10-pt treatment does not mirror the effects of heat shock, the revised document more carefully compares the effect of heat shock and 1,10-pt treatment on RNAP II occupancy. Figures 2a and b show that effect of 1,10-pt on RNAPII occupancy bears little resemblance to the effects of heat shock. Finally, as this reviewer points out, a general stress response triggers a short-term loss of transcription (<15 minutes), after which transcription resumes. As shown in Figure 1a of our manuscript, acetylation does not reappear even after 30 minutes of 1,10-pt treatment, which again argues that the genome-wide loss of acetylation is not due to a general stress response.

Martin et al and their Reply assume that NuA4 (Epl1) binds the vast majority of promoters or gene bodies (thereby justifying the use of 5,542 promoter in their denominator for reporting the fraction of genes losing Epl1 in the presence of 1,10-pt). However, no evidence was presented that shows Epl1 binding to this approximate number of promoters. The assumption of Epl1 detection at most promoters/gene bodies may be statistically no different from background. Therefore, any changes in such background signal (i.e., unsubstantiated Epl1 signal) is meaningless. The statement in Martin's Reply: "the key point is that Epl1 occupancy is largely unaffected by 1,10-pt..." is therefore invalid.

Our original manuscript went through exhaustive peer review by Nature Communications and at no point did any reviewer suggest that a large portion of our data was simply "background", and it seems odd to be addressing this now. If our Epl1 signal was background, then neither mutation of Epl1 nor transcription inhibition should impact this signal. However, disruption of NuA4 by deletion of the C-terminus of Epl1 reduces promoter-bound, but not gene-bound Epl1, and transcription inhibition has the opposite effects (Figures 2a and b of the published manuscript). Further, even promoters that show "substantiated" Epl1 signal following treatment with 1,10-pt, show loss of acetylation, despite increased Epl1 occupancy (Figures 2e and f in the published manuscript), consistent with the transcription-dependence of this PTM.

Both Zencir and Martin agree that there is considerable Epl1 loss (on average) where Epl1 binding can be detected. However, Zencir make the point that gene averaging obscures heterogeneous effects where some gene show loss of binding and others show no change or increased occupancy. This is a common and well know problem with gene averaging, and therefore problematic with the Martin study. For example, Fig 1 of Martin shows a gene average of 5206 genes, but as an average it really reflects only those genes that have the most RNAPII to lose (i.e., highly expressed genes) or the most acetylation to lose, and is not actually representative of all genes.

We invite this reviewer to read our published manuscript. We included heatmaps, showing the log₂ fold-change in RNAPII, histone acetylation, MNase-seq, and Epl1 (Supplementary Figures 3 and 5). These clearly show that decreases in RNAPII, H4K12ac, H3K23ac, and Epl1 occur broadly across all 5206 genes and were not an artifact of plotting the average signal.

The Martin Reply indicates that even genes that gain Epl1 with 1,10-pt treatment lose the corresponding acetylation. Although the effect is modest, as shown in Fig. 2 of the Reply and in prior publications, the result is consistent with Martin's conclusion that HATs can be recruited to at least some promoters without them being active for acetylation. However, the potential for indirect effects does not allow an unequivocal claim. The results are consistent with their model, but is not unequivocal.

While this is a fair statement, we argue that ruling out every conceivable indirect effect in any study is unattainable.

The Martin Reply indicates that Zencir's use of a 400 bp promoter window is flawed. However, the complex being examined here (NuA4) binds to promoters with large NDRs, including RP and other induced genes. This is clearly shown elsewhere when examining other NuA4 subunits (see yeastepigenome.org). As long as these genes are the focus of the analysis, 400 bp is appropriate. Most other genes have small promoters (NFRs), but there is no evidence of Epl1 binding there.

The average size of the NDRs of genes with Epl1 peaks is less than 300 bp. This together with inclusion of a portion of the "+1 nucleosome", makes the definition of promoters by Zencir et al. flawed. Additionally, NuA4 is known to interact with chromatin by binding to 1) transcription activators, 2) H3K4 methylation, 3) H3K36 methylation, and 4) the phosphorylated RNAPII CTD. Thus, one should not expect transcription-factor like binding, with a small number of peaks on the gene-dense yeast genome. Wider NDRs are associated with higher expression in yeast, and the associated increase in NuA4 binding together with the wider NDR, makes Epl1 peaks easier to discern at wide NDRs, but this is not evidence that Epl1 is not bound to promoters within smaller NDRs. Regardless, multiple nucleosome maps are available for defining promoters as the nucleosome depleted regions associated with TSSs, and thus it is no longer acceptable practice to use fixed genomic intervals when defining promoters.

As an aside, the paper behind yeastepigenome.org (PMID 33692541) makes the argument that 2,474 yeast promoters lack sequence-specific TF binding. This rather radical claim, made on the evidence of negative data, suggests that acetylation of ~50% of transcribed yeast genes must occur in the absence of activator targeting, which is consistent with the results of our work.

The Martin Reply states that Zencir Fig. 2D is plotted to locally expand the y-axis that enhances the apparent loss of Epl1. The Martin Reply is correct. All y-axes should start at zero. Further, the analyses in both studies generally suffer from a lack of variance reporting in the metaplots.

As a final note, transcription is a cycle. So, inhibition of one part of the cycle (Pol II trxn) results in loss of histone acetylation. As a cycle, loss of histone acetylation could be temporally downstream of Pol II transcription, but also upstream the next transcription cycle. But even if 1,10-pt were acting directly and exclusively on RNAPII, and causation

of histone acetylation by transcription is established, I don't think this tells us anything definitive about the role of histone acetylation. Transcription might cause histone acetylation and this acetylation might create a chromatin environment that promotes the next round of transcription. So it is a chicken-and-egg question applied to the transcription cycle that is not resolved here.

We agree with the reviewer on this point and note that we wrote something similar in the discussion of our original paper. Indeed, the last line of our Discussion states, "acetylation is a component of a feed forward loop that maintains expression of active genes."

Reviewer #4

Reply to: Pitfalls in the use of phenanthroline to study the causal relationship between promoter nucleosome acetylation and transcription

In their original manuscript, the authors showed that transcription inhibition upon treatment of 1,10 phenanthroline (1,10-pt) had no effect on NuA4 binding to promoter and proposed that histone acetylation is enhanced by RNA pol II transcription rather than promotes RNA pol II recruitment.

Although the authors did not mention a small number of genes showing differential Epl1 binding or those activated by 1,10-pt treatment in the original paper, I generally agree with the data presented in this manuscript and the original paper. In addition, the authors' concerns about the data provided by Albert et al. are also acceptable.

We thank this reviewer for their time and comments.

Reviewer #5

In the Matters Arising, Albert and colleagues raise two important issues that argue against key conclusions from Martin et al. 2021, studying the cause-and-effect relationship between transcription and histone acetylation. The points raised by Albert are:

1. Martin et al. used the drug 1,10 phenanthroline which inhibits RNA Pol II transcription, to show that 15 min of treatment to yeast cells results in the loss of histone acetylation from gene promoters and gene bodies, as well as having effects on the binding of enzymes (HATs) that catalyze histone acetylation. However, Albert raises the point that acute treatment of 1,10 phenanthroline has been shown to affect signaling pathways, transcription factor binding, and induces a heat shock-like stress response. Therefore, the loss of histone acetylation is unlikely to be direct consequence of transcription inhibition by 1,10 phenanthroline.
2. Martin et al. used ChIP-seq to show that 1,10 phenanthroline results in an increase in the binding of a histone H4-specific HAT component Epl1 at a subset of gene

promoters, despite the loss of histone acetylation. Based on this, they argued that promoter-bound HATs are unable to acetylate histones in the absence of transcription. Albert and colleagues reanalyzed the Epl1 ChIP-seq data to show that increased binding is limited to a smaller subset of promoters, while most of the promoters analyzed by Martin et al. actually show a loss of Epl1 binding. Mostly based on this analysis, Albert and colleagues argue that loss of histone acetylation upon 1,10 phenanthroline is rather due to lack of HAT binding or recruitment.

In the rebuttal to the Matter Arising, Martin and Howe argue that careful analysis of the ChIP-seq data with strictly (and more appropriately) defined promoter boundaries show both increase and decrease in Epl1 binding upon 1,10 phenanthroline, while histone acetylation is mainly decreased in both cases. They therefore conclude that loss of acetylation cannot be simply explained by loss of HAT recruitment. Martin and Howe's response to this issue is convincing, and highlights the importance of carefully defining genomic regions for data analysis.

However, this argument over HAT binding is restricted to about 1/10th of all transcriptionally active yeast gene promoters at which Epl1 ChIP-seq shows factor binding in absence of the drug; and is therefore a minor/secondary point. Regardless of Epl1 binding by ChIP-seq, 1,10 phenanthroline results in decreased histone H4 acetylation at promoter-proximal +1 nucleosomes at the majority of yeast genes (not just a subset), as highlighted in the rebuttal. This suggests that Epl1 ChIP-seq cannot be treated as a reliable metric for HAT recruitment or activity. Nevertheless, Martin and Howe focused on mainly addressing this issue about HAT recruitment in their response.

As mentioned in our response to Reviewer #3, NuA4 is proposed to be targeted to transcribed regions via interactions with transcription activators, H3 K4 and K36 methylation, and phosphorylated RNAPII. As such, we do not expect transcription-factor like binding, with a small number of peaks on the gene-dense yeast genome. While wider NDRs make Epl1 peaks more evident, this is not evidence that Epl1 is not bound to promoters within smaller NDRs. Comparison of Figures 2a (all promoters) and 2b (promoters lacking divergent transcription) from our original manuscript underscores this point. Accumulation of Epl1 in a wild-type but not in a Epl1₍₁₋₄₈₅₎ mutant is clearly evident when promoters with divergent transcripts are omitted from the analysis. Thus, we argue that Epl1 ChIP-seq is a reliable indicator of HAT recruitment. The fact that it is not a reliable indicator of HAT activity is one of the key conclusions of this study.

The more important question raised by Albert and colleagues is whether the loss of histone acetylation upon 1,10 phenanthroline is a direct consequence of transcription inhibition. They point out that this can be better addressed with acute depletion of RNAPoIII using an anchor-away or a degron-based approach. Martin et al. have used a RNAPoIII subunit Rpb2 degron in their original study, but limited their analysis to bulk measurements of histone acetylation levels by immunoblotting. Therefore, the genome-wide effects of directly depleting/directly inhibiting RNAPoIII on histone acetylation remains unknown. Martin and Howe point to a more recent study showing similar results in human cells using the TFIIH inhibitor Triptolide (which again does not directly

inhibit RNA PolIII). The rebuttal however demands a more compelling argument in support of a causal role of transcription/RNAPolIII for nucleosomal histone acetylation.

We chose to work with 1,10-pt for the reasons outlined in our response to Reviewer #2, but we would like to point out that a subsequent study using auxin-induced degradation of RNAPII in mammalian cells has generated similar results (PMID 34678064) and thus the genome-wide effects of depleting RNAPII on histone acetylation are known. Also, while this reviewer may argue that triptolide, an inhibitor of the TFIIF helicase, is not a transcription inhibitor, use of this currently preferred method to block transcription initiation in mammalian cells, has reinforced the transcription-dependence of histone acetylation (PMID 35273399). The fact that different groups, using diverse cell systems and distinct approaches to block transcription, see strikingly similar results provides exceptionally strong support for our conclusion that transcription shapes histone acetylation.

In summary, Albert and colleagues have raised very important points to question some key conclusions from a seminal study which addresses a fundamental knowledge-gap in gene regulation. I recommend that this exchange is published with minor revision of the Martin and Howe response to better address the question of causality.